# Measuring Food Insecurity Using the Food Abundance Index: Implications for Economic, Health and Social Well-Being

**DOI:** 10.3390/ijerph17072434

**Published:** 2020-04-03

**Authors:** Audrey Murrell, Ray Jones

**Affiliations:** School of Business and David Berg Center for Ethics and Leadership, University of Pittsburgh, Pittsburgh, PA 15260, USA; rayjones@katz.pitt.edu

**Keywords:** food insecurity, food policy, sustainable food systems, social responsibility, social inequalities, health outcomes

## Abstract

High levels of food insecurity signal the presence of disparities and inequities in local food access, which have been shown to negatively impact the health and well-being of individuals and communities. However, the approaches used to define and measure high food insecurity, also known as a “food desert”, vary widely across research study and intervention methodology. This paper describes the development and validation of a measurement tool called the “Food Abundance Index” (FAI) which is a scorecard for assessing levels of food insecurity across five key dimensions: access, diversity, quality, density, and affordability. A pilot study was conducted to examine levels of food insecurity in order to test the extent to which the FAI can detect food deserts. Nine neighborhoods were selected based on the demographic characteristics of communities shown to be related to food insecurity. Our findings provide evidence that the Food Abundance Index provides a robust measurement tool to assess the extent of food insecurity within a community or neighborhood. Thus, this multidimensional scorecard can be used in future research to detect levels of food insecurity within urban areas and help to bridge the gap between academics, policymakers and practitioners in this important area.

## 1. Introduction

Food insecurity, which reflects the level of access and availability of healthy, nutritious, culturally-appropriate food, is an important indicator of whether people in a community are at risk of hunger, starvation, unhealthy eating, and a host of potential negative consequences, such as malnutrition, obesity, and diabetes, and poor physical, cognitive, and psychological development among children [1]. Measuring high levels of food insecurity also helps in identifying communities that have little or no access to basic foods and where residents suffer from gaps and disparities in obtaining adequate amounts of nutritious foods [2]. There is limited evidence regarding the prevalence of food insecurity among families living in urban areas [3]. Our efforts in this paper are aimed at shedding additional light on the issue of food insecurity and introducing a new measure of food insecurity to address the gaps in existing measurement tools.

Food insecurity has been identified as a significant health and nutrition issue within the United States and globally [4,5,6]. High food insecurity has been associated with key health outcomes, such as risk factors for children’s health and educational outcomes [7], long-term adverse consequences for overall health of children [8], maternal depression and health status [9], mental health conditions [10], subsequent weight gain [11], poor sleep outcomes [12], chronic disease [13], and suicide ideation [14]. Food insecurity has explicit ties to economic conditions, although not exclusively among low income families and households even within food-rich countries such as the United States [15]. This has inspired research calling for immediate action to address food insecurity from a health, economic, and social equality perspectives [16].

The absence or lack of food insecurity represents a troubling social and ethical concern for any society. Food is considered a basic human right—as echoed by the United Nations Declaration of Human Rights, “people have a right to freedom from hunger, and everyone has a right to have access to adequate food” [17,18]. A right to food is a claim that is accepted and acknowledged by individuals and all of society—this is because food, which is essential to all our lives, fulfills a basic biological need (along with water and oxygen), without which an individual would not survive or maintain their well-being. Satisfying this need for food is crucial for an individual to maintain his/her health, dignity and respect. We see high food insecurity as a violation of one’s rights to a healthy and secure life and a denial of the opportunity to realize one’s full potential in society. Consistent with the rights and common good approaches to ethics, access to food is part of society’s ethical and social responsibility, as well as a necessary component of social sustainability. In addition to an individual’s right to basic health and well-being, food insecurity may point to issues of disparity, discrimination, or inequality. For example, in the United States, food insecurity tends to be higher among households with incomes near or below the federal poverty line and black and Hispanic minority groups [19]. Due to a variety of historical, political, and social reasons, households with incomes near or below the poverty line are most likely to be minority households, single parent households, inner city or rural residents, elderly people, and people with disabilities. The incidence of food insecurity among an economically and socially disadvantaged group raises concerns about social justice, economic equality, and discrimination within the current food system [20].

Increasingly, studies have shown that private sector involvement and investment is critical to achieve comprehensive food insecurity [21,22]. It is possible for businesses to address food insecurity and realize significant profits at the same time by adopting a socially responsible business systems approach. This will require challenging the present assumptions of who constitutes the consumer base, understanding the geography of a place, treating food insecure people as potential customers and entrepreneurs, and rethinking the food production/distribution model. A recognition of the link between welfare of individuals and families and the overall sustainability of businesses and communities can help the private sector fulfill its responsibility to promote the welfare of society and safeguard their own interests.

### 1.1. Defining and Measuring Food Insecurity

In order to address the problem of food insecurity, university-based researchers, practitioners, government organizations, community groups, and non-profit research organizations across the country have conducted numerous studies that investigate the prevalence of food insecurity within communities and geographic regions. This has often been done by first identifying areas where vulnerability to food insecurity is high and understanding the factors that constrain people’s ability to obtain enough food. However, methodological approaches vary across researchers and practitioners, which often serve as an obstacle to collaboration or sharing information across these various research endeavors.

Historically, social exclusion, deprivation, retail food prices, household food expenditures, income, and food supply have been studied to understand and document levels of food insecurity. In the early to mid-nineties, the United States Department of Agriculture developed the first national level standard qualitative measure of food insecurity called the “Food Insecurity Core Module” which has been administered every year since 1995. Over the years, researchers and food insecurity experts expanded on the definition of food insecurity and developed new measures and indicators that examine a range of factors known to affect food insecurity. These include:▪Studies on food deserts including geographic measures of food access in food deserts [23].▪Studies examining the retail food environment: availability of supermarkets and other affordable, quality food outlets [24].▪Studies examining the in-store food environment: availability, affordability and quality of healthy nutritious foods and its effect on consumption behaviors [25,26].▪Studies examining the role of race, poverty and neighborhood characteristics in access to food [27].▪Studies examining the health and diet related indicators of food [28,29].▪Studies examining the impact of government food assistance programs on food insecurity [30,31].

While awareness of the problem of food insecurity and the manner in which communities are impacted by low food insecurity has gained national attention as a result of these studies, there are still a lot of gaps and pressing issues that need to be addressed before communities can successfully tackle this problem. In this paper we outline three sources of the divide between academic scholarship and research from practitioners: publication outlets, methodology, and overall research focus.

First, the solutions to address problems of food insecurity need to be based on robust data about the extent of the problem and its consequences. While academics and scholarly researchers have been studying these issues for nearly two decades now, the peer reviewed academic research in which these studies and data are published is often not accessed by practitioners and policymakers. On the other hand, reports on research by practitioners are often published in proprietary reports that come with costs not often within the reach of traditional academic researchers. As a result, the outlets for this work are divergent between academic researchers and practitioners, leading to non-overlapping research streams, which serves as an obstacle for collaboration and knowledge sharing.

Second, scholars and practitioners typically use different metrics and methodologies to study the issue of food insecurity. Researchers frequently use computationally intensive methodologies, but community leaders and practitioners, due to a difficulty of accessing data and limitation of resources and/or expertise, are often unable to undertake rigorous computational approaches to study food insecurity issues. For example, a cursory examination of the titles and contents of 132 peer reviewed articles and practitioner-led published studies reviewed by PolicyLink and the Food Trust that summarized current knowledge/findings about food deserts, food access, and its implications revealed that while a number of studies from both sides examined similar issues, such as the food environment, identification of food deserts, food prices, access, healthy food availability, and so on, the biggest difference was in the methodologies and metrics used [32]. For example, academic researchers examining geographic access to food sources used complex computationally intensive measures such as absolute distance to food source, distance to the nearest food store from a study area centroid, average distance to any mainstream food venue divided by the average distance to a fringe food venue, or distance to the nearest food store from the population-weighted center of each CDB [33]. Practitioner research, besides using average distance to food sources, uses public transit routes, transit schedules, and pedestrian network maps to determine travel time on public transit to the food sources like supermarkets. This difference in metrics and methodology often renders the academic measures/tools inaccessible or useless to practitioners who are interested in using the findings from existing academic tools to develop solutions.

Third, the solutions to food insecurity and food deserts so far have largely been developed and implemented by impacted communities and their leaders. Further examination of the 132 articles mentioned above revealed that practitioner-led studies use a community food systems approach to examine the issue of food insecurity. Such studies have conducted extensive community-wide food assessments which at minimum identified problems and opportunities within the following segments of the food system: (1) agricultural production in the region; (2) food processing and distribution in the region; (3) food consumption; (4) food equity; and (5) food waste and recovery.

These practitioner-led food insecurity studies have placed emphasis on the hunger and nutrition, policy, local agriculture, and community resources aspects of food insecurity. Practitioners have also undertaken research that examined the economic impact of strategies and solutions to eliminate food insecurity implemented like farmers markets, new grocery stores, community supported agriculture programs, food cooperative initiatives, and other business reinvestment strategies. Academic research has yet to catch up with this aspect of applied research to help practitioners, community leaders, and policymakers understand the impact and positive outcomes of comprehensive food retail development initiatives on levels of food insecurity within communities.

In summary, these three sources of division between academic research and work on policy and practice serve as an obstacle to advancing research and solutions in the area of food insecurity. On the one hand, these divides provide critical limitations for academic research on food insecurity to translate into productive community outcomes. Existing tools to measure or track level of food insecurity are often complicated, expensive and not made available to local communities who need reliable information in order to address these negative conditions. On the other hand, these divides result in inconsistencies in methodologies or data that is proprietary and thus not included or viewed as rigorous within traditional academic research. We suggest that providing a common set of tools that can be utilized by both academic and practice researchers would be one step toward closing this divide.

### 1.2. The Food Abundance Index (FAI)

In response to this need, we undertook a pilot project to develop and validate a new measure of food insecurity called the ‘Food Abundance Index’ (FAI). This index has been developed to measure the level of ‘food insecurity’ and uses the presence of ‘food deserts’ as the key metric for assessing potential levels of food insecurity in a specific community. The measure has been designed for broad use by individuals, communities, policymakers, businesses, and corporations and reflects our unique approach to food insecurity as a matter of both business responsibility and social sustainability. The development, design, and pilot testing were done as a collaboration between academic researchers, food industry experts, and food policymakers. Initial steps involved a comprehensive review of existing measures and tools related to food insecurity. Together with food industry and policy experts, a review of the gaps and limitations of existing measures was then conducted. Together with these key stakeholders, a revised set of dimensions was identified that would help to capture the diverse range of existing measures into one comprehensive scorecard.

As mentioned before, the term ‘food deserts’ has been used to characterize community-wide areas that have high rates of food insecurity due to “limited access to affordable and nutritious food, particularly such an area composed of predominantly low income neighborhoods and communities” [34]. Operationally, a food desert is any community or neighborhood-wide area in which residents have little or no access to safe, nutritious food needed to maintain a healthy diet [35]. These economically and geographically disadvantaged areas have the following characteristics:▪They lack healthy food sources like supermarkets and grocery stores, especially within a ten-minute walking distance in urban areas or within a ten-mile radius in rural areas [36].▪They lack variety and choice of healthful food items that meet the nutritional and cultural needs within the community [37].▪They have residents with limited food preparation, food safety, and healthy eating knowledge [38,39].▪They offer poor quality food and limited choices of healthy food [40].▪They have a high concentration of non-healthy food venues, like convenience stores that sell ready-made, boxed, canned or other types of food products as compared to healthy food stores, like grocery and organic stores [41].▪They lack sources of affordable healthy and nutritious food appropriate to the income and purchasing power of the residents [42].

The characteristics mentioned above highlight five related issues pertinent to food deserts that provide the basis for the core dimensions of the FAI scorecard: access, diversity, quality, density, and affordability. People living in food deserts face problems in obtaining adequate amounts of nutritious foods due to gaps and disparities across one or more of these five dimensions, as a result of which people living in food deserts are likely to suffer or be at risk of food insecurity [43,44]. A measurement of the current state of access, diversity, quality, density, and affordability of healthy and nutritious food helps determine whether a ‘food desert’ may exist and reflects the levels of food insecurity that exists within a community. Food deserts, therefore, can serve as an apt indicator of gaps or breakdown in food insecurity in a geographic region. It can also be used to identify where problems related to access, affordability, quality, diversity, and density may put a community at risk of developing food insecurity [45,46]. Our current work utilizes these same five-dimension criteria to assess the presence of food deserts and thereby evaluate potential levels of food insecurity within a specific community or geographic area.

The Food Abundance Index (FAI) provides a useful method for measuring the level of food insecurity, and especially to determine whether a ‘food desert’ may exist within a specific community or neighborhood area. It attempts to combine the strengths of existing measures of food access and availability and examines food insecurity across multiple dimensions: access, diversity, quality, density, and affordability. The FAI uses a scorecard approach similar to the Leadership in Energy and Environmental Design (LEED) certification for communities that awards points for actions across the five dimensions that enhance a community’s level of food insecurity [47]. The scorecard approach allows individuals, communities, and organizations to measure for levels of food insecurity present in a geographic region, identify where problems or gaps may exist, and track changes or progress across multiple dimensions [48]. As mentioned earlier, the FAI measures food availability and accessibility across the five dimensions (access, diversity, quality, density, and affordability) and uses a point-based system to award values across three levels (required, suggested and innovative) in each of five dimensions.

### 1.3. Dimensions and Measurement Criteria

Access. The first dimension of the Food Abundance Index, access is defined as the availability and ease of contact to healthy, nutritious, and balanced food sources. When evaluating for access, the presence of food outlets selling at a minimum fresh produce, meat, and dairy products are taken into consideration. These include mainstream grocery stores (large and small) and alternative food sources such as farmer markets, organic food outlets, and local food source outlets like community supported agriculture, food cooperatives, farm stands, pick your own operations, and so on. But the mere presence of such food sources in itself is not enough if people do not have direct physical access to the grocery store or food outlet. Since minority or economically disadvantaged communities often lack access to supermarkets within a short walking distance people have to rely on public transit to gain access to grocery stores. It was decided that to be classified as accessible, grocery stores must have a serviced bus stop within a five-minute or short walking distance within the designated study area [49].

Diversity. The diversity dimension first analyses the availability of multiple food sources selling healthy nutritious foods within the study area. Diversity of sources is important to look at because areas with high food insecurity usually lack healthful food sources like grocery stores, organic outlets, farmer markets, and so on which typically are outlets that sell nutritious food items like fresh produce, meat, and dairy. This is also an issue of great concern because residents of food insecure areas would have to travel further to get essential food items or may have to depend on the one food outlet (in the absence of diversity) within the neighborhood offering unhealthy food products. Diversity also refers to the presence and availability of a variety of healthy and nutritious food items based on most relevant government sanctioned dietary and nutrition guidelines.

Quality. The duality dimension refers to the presence and availability of appropriately prepared, transported and preserved food that meets the dietary needs of the relevant community. At a minimum everyone should have access to adequately prepared and edible foods. A neighborhood without quality food forces residents to rely on fast food sources or convenience stores or travel outside of their neighborhood to obtain quality food for their family. This FAI dimension determines whether a food outlet sells fresh edible foods and no out-of-date or expired products from each food group of the relevant federal guidelines within the study area. A produce quality rating system is utilized to determine the freshness and appearance of products sold. One can also use existing consumer data on food quality from agencies like the health department to determine the quality of food available within the study area. Food stores can also play a significant role in promoting healthy buying, eating, and consuming habits in their consumers. For example, the presence of healthy dietary intake promotion displays within the store, like signage promoting low calorie foods, fruits and vegetables, or organic or locally grown foods, labels near the food displays giving the nutritional content of the food item, and information on the food guide pyramid or tips for proper food storage and preservation create a high quality in-store food environment. Hence, healthy food promotion is also used as a part of the quality dimension.

Density. Density refers to the proportion of unhealthy food sources to healthy food sources present within the study area relative to health food sources. Food insecure communities typically have an imbalance between unhealthful and healthful food destinations and residents have to rely on venues that sell highly processed foods and foods that are high in salt, fat, and sugar. As a result, food insecure residents are more vulnerable to economic, social, and physical consequences of living in areas with a dearth of outlets that sell nutritious foods. An ideal food environment should have a greater number of food outlets like grocery stores, produce vendors, organic and local food outlets like farmer markets, community supported agriculture, farm stands, and so on, that provide a constant and reliable source for fresh produce, meat/poultry, and dairy items.

Affordability. The final element of the Food Abundance Index, affordability refers to the concentration and availability of healthy and nutritious food sources given the income and purchasing power of residents within the relevant geographic location. Firstly, affordability can be examined by the availability of healthy and nutritious foods at costs less than or equal to the national average cost of purchasing a ‘standard market basket’ of food items [50]. The assessment uses the food list from the relevant federal guidelines, which provides a list of foods items plus condiments needed to prepare a week’s worth of food for an individual or a family. Store surveys can also be conducted to collect information on the lowest price at which food included in the market basket is sold for in the study area. Affordability can then be determined by comparing the cost of purchasing the complete market basket within the study area to the national average cost of purchasing based on the federal or government guidelines for the relevant assessment area [51].

## 2. Materials and Methods

As part of the FAI validation process, a pilot project to test the viability and usability of the FAI was conducted across nine neighborhoods in Pittsburgh, an urban city within the United States made up of ninety distinct neighborhoods. The FAI was first pilot tested on a local scale—a scale that would make the project manageable. There is also presently no study or data that examines the extent of food abundance or food insecurity for the city of Pittsburgh. We examined level of food insecurity across nine Pittsburgh neighborhoods and testing the extent to which the FAI can accurately detect food deserts. The nine neighborhoods selected for this study have all the demographic characteristics of communities that are food deserts: these neighborhoods are among the top ranked neighborhoods for percentage of residents living at 200% poverty level; have large minority populations; high percentages of children under 16 years and adults over 60 years; and low median income, education and employment levels relative to the City of Pittsburgh average (see Table 1 for neighborhood demographic information). We expected the FAI assessment to identify all these neighborhoods, except the ‘Strip District’, as food deserts. The Strip District is a wholesale produce and food district best known for its retail produce and specialty food stores. So, despite having all the characteristics of a food desert, we anticipated the FAI tool would be able to identify this area as not food insecure.

The pilot of the FAI scorecard was developed over a 3-year time period. The first phase involved extensive research on existing measures and methodology related to food insecurity by the research team. After this comprehensive review, a draft FAI scorecard that combined key features of existing tools was then shared with food industry and food policy experts for input and then further refinement. This process followed key steps of critical participatory action research (CPAR) which outlines the use of university research in collaboration with community-based organizations in order to impact and change social policy [52]. This revised version was then field tested by teams of undergraduate business students at a large Midwestern University in the U.S. as part of their service-learning courses. Student teams conducted an inventory of the retail food environment and the in-store food environment across the nine neighborhoods and utilized online resources to collect neighborhood-specific information. These data were analyzed and then used to assess the levels of food insecurity across each of the five FAI dimensions, and a classification was made for each of the specific neighborhoods within the pilot study. Review and approval for conducting this research was submitted to the Institutional Review Board to ensure that all ethical standards for research were followed (Project Internal Case Number: 02907). A summary of demographic characteristics for each of the pilot communities selected for the FAI assessment are provided in Table 1.

## 3. Results

Upon examining the food abundance and insecurity of the nine Pittsburgh’s neighborhoods individually using the Food Abundance Index (FAI), it was evident that these neighborhood communities were not providing adequate access to diverse, healthy, and nutritious food choices to their residents. Eight of the nine-study neighborhoods evaluated fell into the ‘Food Desert’ category, exhibiting highest levels of food insecurity. These results are in line with earlier findings, which found food insecure people to be households with incomes near or below the federal poverty line, children and elderly people, and people with low employment and education levels [44,45].

As expected, the Strip District emerged as a “Food Cluster” as opposed to a food desert. There are no mainstream grocery stores in any of these neighborhoods, and in most of these neighborhoods it was much more likely for residents to go to convenience stores located a few blocks away than to travel to a mainstream grocer for their daily food needs. Even independently owned small corner stores do not succeed in providing access to healthy affordable foods. These neighborhoods are also underserved by alternative food options like farmer markets, community gardens, community support agriculture (CSAs), and so on. This has detrimental impacts on the quality of life of the residents who get a majority of their meals from a fast food or convenient store. A summary of the assessment for each of the neighborhoods across the five dimensions and the resulting category classification is presented in Table 1.

Based on the total score received, each neighborhood’s Food Abundance Index was grouped into one of four possible evaluation or assessment levels. These levels are:▪A ‘Food Desert’ (−5 to 15 points) is the lowest level and most severe level of food insecurity. It is characterized by a severe lack of access to fresh healthy foods and diversity among food outlets and items, extremely poor-quality food, a higher concentration of unhealthy food destinations, and an absence of affordable foods within the study area. Areas designated a ‘Food Desert’ should receive high priority attention to reduce long-term negative impacts.▪A ‘Food Gap’ (16 to 21 points) is at risk for becoming a food desert. Areas or communities at this level may have minimal amounts of access, diversity, quality, density, and affordability but are still providing insufficient amounts for long-term positive outcomes. The ‘Food Gap’ is an early warning sign that unless interventions are put into place, the community could very easily slip towards being a food desert. Food insecurity at this level is still at great risk.▪A ‘Food Cluster’ (22 to 27 points) depicts elevated levels in some of the key dimensions with higher scores in each category. Areas designated at this level have typically achieved at least the minimal on each of the five dimensions. However limited attentions to enhancement across the food insecurity dimensions are present at the ‘Food Cluster’ level. Thus, there are opportunities for innovation at this level.▪A ‘Food Bounty’ (28 to 33 points) entails a strong supply of nutrient rich foodstuffs in a local community, meaning that residents are provided in regard to access, diversity, quality, density, and affordability of the food available in their community. This level also includes innovative approaches to maintain food insecurity.

## 4. Discussion

Based on the initial findings from this pilot study, the Food Abundance Index provides a robust measurement tool to assess the extent of food insecurity within a community or neighborhood. A major contribution of this work is to present a new multi-dimensional measure of food insecurity that strengthens the link between the prevalence of food deserts and food insecurity. We suggest that the adoption and utilization of data produced by the FAI offers a number of policy and management implications to help bring about sustainable social change in the food supply chain.

Firstly, this tool fulfills the need for a reliable, accessible measure of food insecurity that could eliminate critical limitations on academic research to translate into productive community outcomes. Given that this tool was successfully used by undergraduate students in the pilot study, it can similarly be used by community members with little technical skills or expertise to assess food insecurity in their communities. Additionally, through direct participation in the FAI evaluation process community members can feel empowered and motivated to participate in reshaping the current food system and seek solutions to reduce the social and spatial fragmentation in food access and availability that threaten the social sustainability of communities. An analysis of geographic areas and a better understanding of the current state of access, diversity, quality, density, and affordability of foods gathered through tools such as the FAI could provide the basis for action and future direction to policymakers and practitioners and inform evidence-based policies and solutions.

For example, the FAI can help communities understand that strategies like opening a mainstream grocery store in a food desert is just a starting point. Food insecurity is a complex multi-dimensional problem, and a host of strategies that address all aspects of food insecurity will be needed to eliminate this social problem. These include working with stores to add healthy, nutritious foods to their inventory, working with businesses to market healthy foods, providing educational support for healthy consumption, incentivizing people to buy healthy foods, and making alternate sources of healthy foods, like farmers markets, accessible to all, especially low income populations, with acceptance of food stamps at farmer markets. Residents’ food needs can also be used as an economic driver to undertake sustainable policies and market-based enterprises that allow equitable food access and improve food distribution within communities that have been designated as food deserts, i.e., vendor and mobile carts, direct delivery by grocery stores to underserved areas, community gardens, and so on. The scorecard approach used by the FAI to measure food insecurity will prove useful to communities to track improvement in levels of food insecurity over time based on intervention strategies implemented.

Researchers and scholars can use this tool to conduct additional research on the economic, business, and public health impacts of food insecurity as a matter of both social sustainability and responsibility. However, there are some limitations to this study. We do recognize the need to further test, validate, and calibrate the FAI assessment levels to achieve better accuracy in designating a study area a food desert, food gap, food cluster, or food bounty. In the absence of universal standards and multi-dimensional models of food deserts and food insecurity to measure against, the existing assessment and designation criteria will be adjusted to identify and uniformly assign assessment levels across areas exhibiting similar characteristics and indices of deprivation. Also, we have yet to test the tool in communities that do not have the characteristics of food deserts to determine if the tool can successfully identify a community that has high levels of food insecurity as a food cluster of food bounty. The large-scale application of the FAI across the ninety neighborhoods in the City of Pittsburgh must be conducted to fully inform and adjust the FAI assessment levels and the points scoring range. In addition, the cross-validation of the FAI tool with other measures of food insecurity should be conducted to further assess the impact and fidelity of this innovative scorecard. This poses an important limitation of the current data as well as the opportunity for future research.

## 5. Conclusions

Identifying food-insecure areas is the first steps in helping communities reach the ultimate goal of building an ethical and equitable food system. The information gathered through tools such as the Food Abundance Index about the current state of access to and availability of healthy, affordable foods could provide the basis for greater awareness and understanding of the issue and inspire change, action, and future directions for academics and practitioners. Thus, our work in developing and distributing the FAI tool reflects our attempt to bridge the gap and expand the boundaries between academia and practitioners using robust methodology.

The tool has been designed to be accessible and easy to use by practitioners, especially those with limited financial and research resources, to study the problems of food insecurity, identify shortcoming in the local food system, and also develop evidence-based solutions/policies. In addition, the FAI tool considers food insecurity across multiple dimensions and provides a complete picture of food insecurity in a community as related to access, diversity, quality, density and affordability. As a measurement tool, the FAI can not only assess whether a food desert exists within a particular community but can also be used to identify where gaps or problems may exist that may cause a breakdown in food insecurity in the future. Lastly, the FAI uses a scorecard approach to measure for different levels of food insecurity, track improvement in levels of food insecurity over time, and provide a benchmark that practitioners can use in addressing the issue of food insecurity.

We argue that the adoption and utilization of data produced by the FAI offers a number of potential contribution and benefits. Analysis of geographic areas using the FAI will help lay the foundation for reducing communities’ food insecurities and improving economic status, health, and overall well-being. We expect use of the FAI to also stimulate additional academic research by providing researchers with a comprehensive tool to measure levels of food insecurity, as well as outcomes and the impact of change efforts. Lastly, we anticipate use of the FAI tool to facilitate collaboration, knowledge sharing, and dialog between academic and community-based researchers based on the common language, methodology, and focus provided by this new approach to understand the dimensions and impact of food insecurity as a key social issue.

## Figures and Tables

**Table 1 ijerph-17-02434-t001:** Neighborhood demographics and Food Abundance Index (FAI) results.

Neighborhood	Race (%White/%Black)	Median HOUSEHOLD Income (USD)	% People Over 60	% Family/Individual Below Poverty Level	% Employed	FAI Results (Category)
Glenn Hazel	23.9%/72.4%	$13,824	40.5%	50.7%/60.5%	26.3%	Food Desert
Upper Hill District	71.9%/21.2%	$69,091	24.8%	6.0% /33.8%	19.2%	Food Desert
Uptown (Bluffs)	72.2%/22.6%	$31,667	21%	23.0%/28.5%	52.7%	Food Desert
Manchester	12.1%/85.5%	$33,712	18.4%	15.0%/23.1%	47.7%	Food Desert
Middle Hill District	2.1%/96.8%	$23,009	27.1%	24.4%/33.5%	42.4%	Food Desert
Lawrenceville	83.8%/12%	$24,138	25.5%	19.3%/21.1%	46.6%	Food Desert
Polish Hill	3%/60.9%	$16,029	13.6%	36.2%/37.5%	55.2%	Food Desert
Downtown	10.4%/86.8%	$31,417	26.7%	16.4%/17.1%	51.3%	Food Desert
Strip District	78.4%/13.3%	$16,964	4.8%	33.6%/58.6%	36.9%	Food Cluster

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
