# Peer review of "Measuring Food Insecurity Using the Food Abundance Index: Implications for Economic, Health and Social Well-Being"

_ijerph, 2020, doi:10.3390/ijerph17072434_

Round 1

Reviewer 1 Report

The authors have clearly demonstrated their deep knowledge of this subject matter and it was a pleasure reviewing the article.

My only concern is whether the FAI was developed with consultation from practitioners and other key stakeholders (e.g. policymakers, food industry experts, etc). If this was developed solely by academic researchers, I question how differential this will be compared to the other academic research as described in the third page of your manuscript.

Author Response

A further clarification of the process to develop the Food Abundance Index that engaged academic researchers, policymakers and food industry experts has been added (see page 4).  Within the section on the Food Abundance Index, the following information is now included:

"The development, design and pilot testing were done as a collaboration between academic researchers, food industry experts and food policymakers. Initial steps involved a comprehensive review of existing measures and tools related to food insecurity. Together with food industry and policy experts, a review of the gaps and limitations of existing measures was then conducted. Together with these key stakeholders a revised set of dimensions was identified that would help to capture the diverse range of existing measures into one comprehensive scorecard."

Reviewer 2 Report

  • The topic is not descriptive enough; for example, it does not provide the context of the study (Where was the study done)?
  • The abstract is too brief, should include some methodology and key findings (briefly), including a recommendation.
  • There is no clear distinction between food security and insecurity concepts in the article, seems like the two are used interchangeably. A clear distinction can be provided in the introduction.
  • My concern in the methodology is that the study relies on a single tool (FAI) to measure food (In)security, it would be better to have another tool to supplement the FAI.
  • No ethical compliance is mentioned throughout the article. Who issued ethical clearance for the study?
  • There is no mention throughout the article as to when the study was conducted?
  • The results are too brief, not adequately presented.
  • The Authors must check consistency in use on some terms/ words throughout the article.
  • The referencing style used does not seem to follow the journal style (in-text references appear like footnotes).

Author Response

See attached authors responses to Reviewer #2 comments.

Round 2

Reviewer 2 Report

The authors have attended to comments/ suggestions, the article may be considered for publication.